# Tyrosinase Inhibition and Kinetic Details of Puerol A Having But-2-Enolide Structure from *Amorpha fruticosa*

**DOI:** 10.3390/molecules25102344

**Published:** 2020-05-18

**Authors:** Jeong Ho Kim, Da Hyun Jang, Ki Won Lee, Kwang Dong Kim, Abdul Bari Shah, Kamila Zhumanova, Ki Hun Park

**Affiliations:** 1Division of Applied Life Science (BK21 plus), IALS, Gyeongsang National University, Jinju 52828, Korea; rwg2610@gnu.ac.kr (J.H.K.); ekgus6383@naver.com (D.H.J.); abs_korea28@gnu.ac.kr (A.B.S.); zhumanovakamila18@gmail.com (K.Z.); 2Division of Applied Life Science (BK21 plus), PMBBRC, Gyeongsang National University, Jinju 52828, Korea; leemaskup@naver.com (K.W.L.); kdkim88@gnu.ac.kr (K.D.K.)

**Keywords:** *Amorpha fruticosa*, puerol A, tyrosinase, binding affinity, anti-pigmentation

## Abstract

Puerol A (**1**) from *Amorpha fruticosa* showed highly potent inhibition against both monophenolase (IC_50_ = 2.2 μM) and diphenolase (IC_50_ = 3.8 μM) of tyrosinase. We tried to obtain a full story of enzyme inhibitory behavior for inhibitor **1** because the butenolide skeleton has never been reported as a tyrosinase inhibitor. Puerol A was proved as a reversible, competitive, simple slow-binding inhibitor, according to the respective parameters; *k*_3_ = 0.0279 μM^−1^ min^−1^ and *k*_4_ = 0.003 min^−1^. A longer lag-phase and a reduced static-state activity of the enzyme explained that puerol A had a tight formation of the complex with E_met_. Dose-dependent inhibition was also confirmed by high-performance liquid chromatography (HPLC) analysis using *N*-acetyl-l-tyrosine as a substrate, which was completely inhibited at 20 μM. A high binding affinity of **1** to tyrosinase was confirmed by fluorescence quenching analysis. Moreover, puerol A decreased melanin content in the B16 melanoma cell dose-dependently with an IC_50_ of 11.4 μM.

## 1. Introduction

Tyrosinase (EC 1.14.18.1) is the type-3 metalloenzyme, which is most highly related with the production of melanin. Melanin production in the living organism is a natural adaption to the outside environment for protecting the skin from ultraviolet damage and reactive oxygen species (ROS). The skin hyperpigmentation caused by excessive melanin is a pressing issue in the cosmetic market, together with wrinkles [1]. On the other hand, tyrosinase also attributes to neuromelanin production from dopamine, which may lead to dopamine neurotoxicity and further neurodegeneration involved with Parkinson’s disease [2]. Most of the strategies of controlling melanin production have focused on the regulation of tyrosinase activity. Tyrosinase catalyzes two steps with monophenolase and diphenolase at the same active site as follows; the hydroxylation of l-tyrosine to l-3,4-dihydroxyphenylalanine (l-DOPA) and the oxidation of l-DOPA to dopaquinone. The subsequent steps from dopaquinone to melanin are extremely rapid and are non-enzyme catalyzed processes [3]. Antipigmentation could be estimated by inhibitory potencies of both monophenolase and diphenolase. Mostly, the inhibitors of tyrosinase are derived from phenolic compounds because they have structural similarity with the substrate, l-tyrosine. The representative inhibitors are flavonoids, hydroxy stilbenes, cinnamic acid derivatives, anthraquinone, and simple phenols [4]. The current study has mainly focused on exploring a lead structure with a new skeleton having high inhibitory potency.

In our continuing search for tyrosinase inhibitors of natural origin [5], the methanol extract of the roots of *Amorpha fruticosa* was confirmed to inhibit tyrosinase. This plant belongs to the *Leguminosae* family and is distributed throughout Korea and China. The root part has been used traditionally in Chinese folk medicine against ambustion, carbuncle, and eczema [6]. Previous studies reported that the main secondary metabolites are rotenoids [7], prenylated flavanones [8], isoflavones [9] and stilbenes [10]. Many of them exhibited cytotoxic, antidiabetic, and antimicrobial activities [11,12,13]. Moreover, the prenylated flavanones exhibited significant inhibition against bacterial neuraminidase [14]. In this study, we isolated several phenolic compounds from the methanol extract of *A. fruticosa* together with tyrosinase inhibitory but-2-enolides. But-2-enolides have never been reported as a skeleton for tyrosinase inhibition so far. Chemical structures were identified using spectroscopic methods and compared to previous data. The details of the inhibitory mechanism, including slow-binding behavior were ascertained using Lineweaver–Burk and Dixon plots. The high-performance liquid chromatography (HPLC) analysis method was also carried out to estimate the inhibitory potency. The binding affinity between the inhibitor and enzyme was measured by the fluorescence quenching method. Moreover, the isolated inhibitors ware applied to a B16 melanoma cell to measure the anti-pigmentation effect.

## 2. Results and Discussion

### 2.1. Isolation of But-2-Enolides

In the course of exploring a lead structure for tyrosinase inhibition, we isolated two but-2-enolides (Figure 1) from the methanol extract of the roots of *A. fruticosa*. Compound **1** was obtained as a yellow powder having the molecular formula C_17_H_14_O_5_ established by high-resolution electrospray ionization mass spectrometry (HRESIMS) (*m*/*z* 299.0843 [M]^+^, calcd 299.0841). The but-2-enolide functionality of **1** was deduced by α,β-unsaturated carbonyl C1 (δ_C_ 175.7) and α−position H2 (δ_H_ 6.13), which had heteronuclear multiple bond correlations (HMBC) with C3 (δ_C_ 167.1) and oxygenated carbon C4 (δ_C_ 84.4) (Appendix A). The compound **2** was elucidated as glucoside of **1**. The purity of compounds **1** and **2** was evaluated by HPLC analysis (Appendix A). The spectroscopic data of the isolated compounds (**1** and **2**) accord with those previously published for puerol A (**1**) and kuzubutenolide A (**2**) [15,16].

### 2.2. Tyrosinase Inhibition

Numerous tyrosinase inhibitors have been isolated from the plant source. The flavonoids are the most representative class because of their structural similarity with a substrate, l-tyrosine [17]. The exploring tyrosinase inhibitor has been extended to stilbenes, phenylpropanoic acids, anthraquinone, and lignin to take out better lead structure. Therein, from the very beginning, our results demonstrating in Figure 2a show that compound **1** inhibits the monophenolase activity of tyrosinase dose-dependently with ranges of 0.5–8.0 μM. It inhibited both monophenolase (l-tyrosine → l-DOPA) and diphenolase (l-DOPA → dopaquinone) with IC_50_ values of 2.2 μM and 3.8 μM, respectively (Table 1). Whereas compound **2**, which is the glucoside of **1,** showed relatively low inhibitory activity to monophenolase (IC_50_ = 1.5 mM) and diphenolase (IC_50_ = 1.9 mM). Specifically, the advantage of compound **1** is inhibiting both monophenolase and diphenolase activities with similar potencies. Furthermore, it is the first report to highlight that the but-2-enolide skeleton has a great potential for tyrosinase inhibition.

The reversibility of puerol A (**1**) was proved by Figure 2b, which was plotted a change of the initial velocity according to tyrosinase and puerol A (**1**) concentrations. The lines passing through the same origin indicated that puerol A is a reversible inhibitor to tyrosinase.

The inhibitory behaviors of puerol A (**1**) were estimated using Lineweaver–Burk and Dixon plots. As shown in Figure 2c, puerol A (**1**) inhibited monophenolase activity of tyrosinase competitively, because of a common intercept on the y-axis with V_max_ of 5.90 OD/min by increasing the concentration of **1**. The Dixon plot was obtained by plotting 1/V vs. [I] (The concentration of inhibitor) with varying concentrations of substrate for the *K*_i_ value to be 0.87 μM. As shown in Figure 2e, the kinetic plot indicates that compound **1** also has a competitive inhibition mode of action to diphenolase. A family of straight lines of the Lineweaver–Burk plots has the same y-axis intercept with a *V*_max_ of 33.90 OD/min. The *K*_i_ value of **1** to diphenolase was 1.95 μM by the Dixon plot, as shown in Figure 2f.

To investigate the inhibitory mechanism further, a time-dependent inhibition experiment was disclosed because compound **1** was a competitive inhibitor of tyrosinase. The first step was to measure the initial velocities for the substrate (l-tyrosine) oxidation at 2.0 μM of the inhibitor concentration by preincubation time. A progressive loss in residual enzyme activity was observed by the preincubation time, which was assigned as a 5 min interval until 30 min (Figure 3a). Enzyme activity was maintained during 60 min preincubation as shown in the Figure 3a inset. An increase of preincubation time led to the decrease in the initial velocity based on residual enzyme activity. Both, the initial velocity (*v*_i_) and steady-state rate (*v*_ss_) were reduced by the increasing concentration of **1** (Figure 3a). Thus, puerol A (**1**) had a slow-binding characteristic of tyrosinase, which could bind to the enzyme tightly. The *K*_obs_ values on the concentration of inhibitor were taken from Equations (2) and (3) to give a straight pattern at the plot of *K*_obs_ vs. the inhibitor concentration (Figure 3b inset). This indicated compound **1** is a simple, reversible, slow-binding inhibitor. Furthermore, the fitting of Equations (4) and (5) to the results gave the following kinetic parameters; *k*_3_ = 0.0279 μM^−1^ min^−1^, *k*_4_ = 0.003 min^−1^, and $K_{i}^{app}$ = 0.1075 μM.

The active site of tyrosinase has three different states such as E_oxy_, E_met_, and E_deoxy_ during the oxidation processes. The E_oxy_ form can work on the oxidation of both substrates of l-tyrosine and l-DOPA, but the E_met_ form can work on l-DOPA oxidation [18]. Thus, the formation of E_met_∙I complex might delay the enzyme reaction to give lag time. The lag time was obtained from the kinetics course of the substrate (l-tyrosine) oxidation by the increasing of inhibitor concentrations (Figure 4a). The lag phase in the absence of the inhibitor was measured as 60 s, but the lag time was prolonged with the dose-dependence of the inhibitor **1** concentration, in the range of 0.5–8.0 μM up to 600 s of lag phase. The steady-state rate was determined by the extrapolation curve to the abscissa, as shown in Figure 4b. With the different tendencies of the lag time, the steady-state rate (*v*_ss_) was decreased by increasing the inhibitor **1** concentration. The results indicated that inhibitor **1** bound to E_met_ form effectively to decrease the steady-state rate.

### 2.3. HPLC Analysis of Tyrosinase Inhibition

The above-mentioned results of tyrosinase inhibition were examined for the change of melanin color at 475 nm. In addition, we tried the double verification of tyrosinase inhibition for puerol A by using different substrates and UV absorbance at 275 nm through HPLC analysis. The procedure was based on the use of *N*-acetyl-l-tyrosine instead of l-tyrosine as a substrate for tyrosinase. Tyrosinase could oxidize *N*-acetyl-l-tyrosine up to *N*-acetyl dopaquinone via the intermediate, *N*-acetyl-l-DOPA owing to *N*-acetyl group blocking a further cyclization to dopachrome [17]. Such wise, measuring of the peak a (*N*-acetyl-l-tyrosine) degradation using HPLC at 275 nm allows the detection of inhibitory potency. Figure 5a indicated that *N*-acetyl-l-tyrosine (peak a) was oxidized time-dependently by tyrosinase and converted to other oxidized compounds completely in 60 min. HPLC analysis serves as evidence that puerol A inhibited oxidation functions of monophenolase and diphenolase of tyrosinase at low concentrations in comparison with other inhibitors. The oxidation of *N*-acetyl-l-tyrosine was blocked by the dose-dependent concentration of puerol A, such as 56% at 5 μM, 86% at 10 μM, and 95% at 20 μM (Figure 5b–d). Thus, puerol A was verified as a potent tyrosinase inhibitor by another method, i.e., HPLC analysis.

### 2.4. Binding Affinity to Tyrosinase

We also estimated the binding affinity of compound **1** to tyrosinase. Tyrosinase enzyme has the fluorescent residues, seven tryptophan residues (Trp 106, 124, 141, 195, 227, 350, and 386) and nine tyrosine residues (Tyr 51, 65, 78, 140, 165, 236, 311, 314, and 382) [19]. The intrinsic fluorescence of tyrosinase might be quenched as a function of the inhibitor. The binding affinity was measured by a fluorescence quenching effect. The measurement was carried out at the emission wavelength (300–400 nm), where none of the other components showed any significant emission. The Stern–Volmer plots demonstrated that the fluorescence intensity of tyrosinase was reduced according to the concentration of **1** from 2.5 μM to 20 μM. The quenching tendency was steady in the way of inhibitory potencies based on the comparison of **1** (IC_50_ = 2.2 μM) and **2** (IC_50_ = 1.8 mM), as shown in Figure 6b (see Appendix A for compound **2**). Using the Stern–Volmer equation, the *K*_SV_ value was estimated as 1.45 × 10^5^ L·mol^−1^. From the plots of the linear Equation (7), *K*_A_ and n were calculated and the value of n approximates to one, showing that only a single binding site exists in tyrosinase for inhibitor **1**. The results proved that the potent inhibition of **1** arises from an effective binding affinity to tyrosinase. 

### 2.5. Antipigmentation Activity In Vitro

The isolated puerol A (**1**) was examined for its ability to suppress pigment formation in B16 melanoma cells which was stimulated through the production of melanin by α-melanocyte-stimulating hormone (α-MSH). The inhibitor concentrations were adjusted to be non-toxic to B16 melanoma cells (Appendix A). Puerol A showed a significant reduction of melanin content dose-dependently from 20 μM to 80 μM concentrations (Figure 7a). The inhibitory potency was estimated as 11.4 μM of IC_50_. Tyrosinase inhibition in B16 melanoma cells was also estimated with the protection of l-DOPA oxidation, as shown in Figure 7b. The l-DOPA oxidation was inhibited by puerol A with 23.9 μM of IC_50_ as presented in Table 2.

## 3. Materials and Methods

### 3.1. General Experimental Procedures

Open column chromatography was carried out using MCI GEL CHP20P (75–150 μm, Merck, Kenilworth, NJ, USA). Preparative recycling medium pressure liquid chromatography (MPLC) and HPLC were conducted on an LC-9130G NEXT (Jai Co., Ltd., Tokyo, Japan) using AQ C18 (S-10 μm, 12 nm, YMC, Kyoto, Japan) and Acclaim Polar Advantage II C-18 (S-5 μm, 12nm, Thermo Fisher Scientific, Waltham, MA, USA). One-dimensional, as well as 2D NMR spectra, were recorded by a Bruker AM500 spectrometer (Billerica, MA, USA), using either methanol-*d*_4_ or acetone-*d*_6_ as solvent and tetramethylsilane (TMS) as an internal standard. UV spectra were obtained using a DU650 spectrophotometer (Beckman Coulter, Brea, CA, USA). HRESIMS were conducted using Vion (Waters, Milford, MA, USA). Enzymatic assays were implemented using a SpectraMax M3 Multi−Mode microplate reader (Molecular Devices, San Jose, CA, USA). The specific rotation was estimated using a P−2000 Digital Polarimeter (JASCO, Tokyo, Japan). All chemicals for analysis were of first grade. The roots of *A. fruticosa* were harvested at Naedong in Jinju, South Korea, in June 2017.

### 3.2. Extraction and Isolation

The barks (0.5 kg) from the dried roots of *A. fruticosa* was extracted with methanol (10 L × 3) for 10 days at room temperature. The accumulated filtrate was evaporated to yield a dark brown residue (24 g), which was suspended in H_2_O (0.5 L) and further portioned with hexane (1 L × 3) and ethyl acetate (1 L × 3). The ethyl acetate fraction (7.6 g) was subjected to MCI gel (200 g) using a gradient of water to methanol (10:1 to 1:10, *v*/*v*), which gave 10 fractions (A–J). Tyrosinase inhibitory fraction D-E (2.1 g) was fractionated using MPLC with reversed silica gel CC (250mm × 30.0 mm, S-10 μm, 12 nm, YMC) and eluted with a gradient of MeOH in H_2_O (0 to 100%, *v*/*v*) at a rate of 15 mL/min, to give rise to fifty subfractions (D1–D50). Sub-fractions D15–18 (72.7 mg), enriched with compound **1,** were purified by recycling HPLC (250 mm × 30.0 mm, S-5 μm, 12 nm, Thermo Fisher Scientific) having isocratic elution with H_2_O: ACN (1:1) to give rise to compound **1** (32 mg). Sub-fractions D23–27 (81.2 mg) provided compound **2** (45 mg) under the equal recycling HPLC condition.

#### 3.2.1. Puerol A (Compound **1**)

Pale yellow powder. UV (MeOH) λ_max_ (log *ε*) 290 (3.95) 319 (4.01). [α]_D_ + 131.6° (MeOH; c 0.10). HRESIMS [M]^+^ 298.0841 (calcd. for C_17_H_14_O_5_, 298.0841). ^1^ H-NMR (500 MHz, MeOH-*d*_4_) δ 2.80 (1H, dd, *J* = 6.0, 14.5 Hz, H-4a), 3.25 (1H, dd, *J* = 3.5, 14.5 Hz, H′-4a), 5.95 (1H, ddd, *J* = 1.1, 3.5, 5.8 Hz, H-4), 6.13 (1H, d, *J* = 1.1 Hz, H-2), 6.44 (1H, d, *J* = 1.5 Hz, H-3″), 6.45 (1H, d, *J* = 2.3 Hz, H-5″), 6.62 (2H, d, *J* = 8.5 Hz, H-3′, H-5′), 6.86 (2H, d, *J* = 8.5 Hz, H-2′, H-6′), 7.30 (1H, dd, *J* = 1.6, 7.4 Hz, H-6″) (see Appendix A).

#### 3.2.2. Kuzubutenolide A (Compound **2**)

Pale yellow oil. UV (MeOH) λ_max_ (log *ε*) 289 (4.01), 310 (4.06). [α]_D_ +66.2° (DMSO; c 0.10). HRESIMS [M]^+^ 460.1368 (calcd. for C_23_H_24_O_10_, 460.1369). ^1^ H-NMR (500 MHz, Acetone-*d*_6_) δ 2.79 (1H, dd, *J* = 6.2, 14.7 Hz, H-4a), 3.19 (1H, dd, *J* = 3.7, 14.6 Hz, H′-4a), 3.52 (1H, m, H-4‴), 3.58 (1H, m, H-3‴), 3.59 (1H, m, H-5‴), 3.60 (1H, m, H-2‴), 3.93 (1H, m, H-6‴), 5.18 (1H, t, *J* = 7.1 Hz, H-1‴), 5.95 (1H, ddd, *J* = 1.1, 3.5, 5.8 Hz, H-4), 6.13 (1H, d, *J* = 1.1 Hz, H-2), 6.44 (1H, d, *J* = 1.5 Hz, H-3″), 6.45 (1H, d, *J* = 2.3 Hz, H-5″), 6.62 (2H, d, *J* = 8.5 Hz, H-3′, H-5′), 6.86 (2H, d, *J* = 8.5 Hz, H-2′, H-6′), 7.30 (1H, dd, *J* = 1.6, 7.4 Hz, H-6″) (see Appendix A).

### 3.3. Inhibitory Effects Against Tyrosinase

Mushroom tyrosinase (EC 1. 14. 18. 1) (Sigma-Aldrich Co., St. Louis, MO, USA) activity was performed in accordance with the previously described method with subtle modification, using l-tyrosine and l-3,4-dihydroxylphenylalanine (l-DOPA) for monophenolase and diphenolase as substrates, respectively [20]. Two hundred sixty units/mL of the tyrosinase stock solution was prepared at optimum pH 6.8 (0.05 M phosphate buffer). The inhibitors were initially dissolved in DMSO and diluted to several concentrations. In brief, in a 96-well plate, 5 μL of inhibitor and 20 μL of 1.8 mM of l-tyrosine or 3.6 mM of l-DOPA were added as a substrate in the aforementioned buffer (165 μL). Then, 10 μL of the enzyme was added to the mixture. Subsequent absorbance of DOPA chrome formation was recorded at 475 nm by means of an UV-Vis spectrophotometer (SpectraMax M3, Molecular Devices, San Jose, CA, USA) in a 96-well microplate at 30 °C. Three separate determinations were conducted for each assay. The inhibitory activity of isolated compounds was estimated by the concentration, which inhibited 50% of enzyme activity (IC_50_). The percentage of inhibition was determined by Equation (1) as follows:(1)$$\mathrm{Activity}\left(\%\right){=100\;[1/(1+([I]/IC}_{50}))]$$

The characteristic of tyrosinase inhibition was represented by Lineweaver−Burk double-reciprocal-plot and Dixon plot at specified concentrations of the substrates and inhibitors, respectively.

### 3.4. Time-Dependent Assays and Progress Curves

The assays were carried out using tyrosinase (260 units) and l-tyrosine as a substrate of monophenolase in 0.25 M phosphate buffer (pH 6.8) at 30 °C. The enzyme activity was recorded sequentially for 30 min by a UV spectrophotometer. To evaluate the time-dependent inhibition and the kinetic parameters of tyrosinase, progress curves with 30 s intervals were measured at different concentrations of the inhibitor and substrate. The data were through the use of Sigma Plot (SPCC Inc., Chicago, IL, USA) as a nonlinear regression program to provide the respective parameters for each curve; A (absorbance at 475 nm), *v*_i_ (initial rate), *v*_ss_ (steady-state rate), *k*_obs_ (apparent first-order rate constant for the transition from *v*_i_ to *v*_ss_), and $K_{i}^{app}$ (apparent *K*_i_) in accordance with the following Equations (2)–(5) [21].
(2)$${A=v}_{\mathrm{ss}}t+\left(v_{i}{\;-\;v}_{\mathrm{ss}}\right)\left[1\;-\exp\left({-k}_{\mathrm{obs}}t\right)\right]{/k}_{\mathrm{obs}}$$
(3)$${v/v}_{0}{=\exp\;(-k}_{\mathrm{obs}}t)$$
(4)$$k_{\mathrm{obs}}{=k}_{4}{\;(1+[I]/K}_{i}^{\mathrm{app}})$$
(5)$$K_{i}^{\mathrm{app}}=\frac{k_{4}}{k_{3}}$$

The value of ${-k}_{\mathrm{obs}}$ was estimated by a slope of the straight line from the plot of the natural log of the residual enzyme activity vs. the preincubation time. The plot of *k*_obs_ vs. the concentration of inhibitors gave the values of *k*_3_, *k*_4_, and $K_{i}^{app}$ on the basis of Morrison and Walsh’s method.

### 3.5. HPLC Analysis of Tyrosinase Inhibition

The HPLC analysis of tyrosinase inhibition was carried out by giving minor alterations in the previous procedure [22] with *N*-acetyl-l-tyrosine as a substrate. The 4.5 mM of the substrate was prepared in the 50 mM phosphate buffer (pH 6.8). Then, 670 units/mL tyrosinase (Sigma-Aldrich Co., St. Louis, MO, USA) solution and several concentrations of the inhibitor solution were arranged. The *N*-acetyl-L-tyrosine was observed on 275 nm with 4.6 min of tR by eluting the mobile phase consisting of A (H_2_O) and B (acetonitrile, ACN), with the following isocratic condition: 0–8 min and A/B = 90:10 (*v/v*). After 20 μL of the sample solution was subjected to the HPLC (Pack ODS-AQ C18 column, YMC) system, the reactions were measured every 15 min. All procedures were carried out at room temperature.

### 3.6. Fluorescence Quenching Measurements

To measure fluorescence from the tyrosinase enzyme, 10 μL of 260 unit/mL enzyme solution with 165 μL of phosphate buffer (0.05 M) were added into 96-well black immunoplates. Afterward, 5 μL of various concentrations (2.5–20 μM) of inhibitor was added into each well. All fluorescence spectra were obtained by a SpectraMax M3 spectrophotometer (Molecular Devices, San Jose, CA, USA). The spectra were measured from 300 to 400 nm with emission slits regulated to 2.0 nm, and the excitation wavelength was 265 nm. The Stern–Volmer quenching constant (*K*_SV_) was estimated using Equation (6). All experiments were carried out in triplicate, and the mean values were calculated [23].
(6)$$F_{0\;}{-\;F=1+K}_{SV}\left[Q\right]$$
where *F_0_* and *F* are the fluorescence intensities in the absence and presence of quencher (Q). *K*_SV_ is the Stern–Volmer quenching constant [L M^−1^]. For static quenching, the correlation between the fluorescence intensity and the concentration of the quencher for the series of reactions can be expressed by Equation (7).
(7)$${\log[(F}_{0}{\;-\;F)/F]=\log K}_{A}{+\mathrm{nlog}[Q]}_{f}$$

*F_0_* and *F* are the fluorescence intensities in the absence and presence of inhibitor; *K*_A_ is the binding constant; *n* is the number of binding sites of the enzyme; Q_f_ is the concentration of inhibitor [24].

### 3.7. Cell Culture

The B16F10 mouse melanoma cell (American Type of Culture Collection, VA, USA) were cultured as previously reported [25]. Briefly, the cells were incubated in DMEM containing 10% fetal bovine serum and 1% penicillin/streptomycin (Sigma–Aldrich, St. Louis, MO, USA) at 37 °C in a 5% CO_2_ humidified incubator.

### 3.8. Cell Viability Assay

The cell viability assay of the B16F10 mouse melanoma cells were conducted with subtle variation in previous methods [26]. The cells were incubated in 96-well plates overnight. Then, the cells were treated with different concentrations of compound **1** and incubated for 24 h. After incubation, 5 μg/mL of MTT solution was added to the each well and the cells were incubated for 3 h. After the medium was removed, the cells were treated with dimethyl sulfoxide (DMSO) and cultured at room temperature for 20 min. The absorbance at 595 nm was measured on a Bio-Rad microplate reader (Hercules, CA, USA).

### 3.9. Measurement of Melanin Content

For the analysis of melanin content accumulated in the cytosol, B16F10 mouse melanoma cells were seeded on to six-well plates at a density of 1 × 10^5^ cells per well, and cultured for 24 h. Then, the cells were treated with 1 μM of α-MSH (Sigma–Aldrich, St. Louis, MO, USA) in the presence or absence of different concentrations of compound **1**. Thereafter, the cells were collected using trypsin-EDTA (Lonza, Walkersville, MD, USA) and dissolved in 1 N NaOH, including 10% DMSO at 65 °C for 24 h. The melanin content was determined at 415 nm by a Bio-Rad microplate reader [27,28].

### 3.10. Measurement of l-DOPA Oxidation

B16F10 mouse melanoma cells were seeded on to 6-well plates at a density of 1 × 10^5^ cells per well and cultured for 24 h. Then, the cells were treated with α-MSH (1 μM) in the presence or absence of different concentrations of compound **1**, followed by incubation for 48 h. The incubated cells were collected and dissolved with 1% Triton X-100 solution for 1 h on ice. Then, the cells were cultured for 4 h with 100 μL (2 mg/mL) l-DOPA in a 5% CO_2_ humidifying incubator at 37 °C condition. After that, by using a microplate reader (Bio-Rad), absorbance at 490 nm was measured [25].

### 3.11. Statistical Analysis

All experiments were carried out at least thrice. The data was conducted to variance analysis using Sigma Plot (version 10.0, Systat Software, Inc., San Jose, CA, USA). The value of *p* < 0.05 was regarded to be a significant difference.

## 4. Conclusions

In this study, we disclosed that but-2-enolide had great potential for tyrosinase inhibition. Puerol A (**1**) inhibited both monophenolase (IC_50_ = 2.2 μM) and diphenolase (IC_50_ = 3.8 μM) potentially. Inhibitory behavior to the enzyme was proved by Lineweaver-Burk plot, HPLC analysis, and fluorescence quenching. It was proved that puerol A was competitive inhibitor and reversible, simple slow-binding inhibitor according to respective parameters; *k*_3_ = 0.0279 μM^−1^ min^−1^, *k*_4_ = 0.003 min^−1^ and $K_{i}^{app}$ = 0.1075 μM. The anti-pigmentation effect of puerol A was also observed on the B16 melanoma cell assay with 11.4 μM of IC_50_.

## Figures and Tables

**Figure 1 molecules-25-02344-f001:**
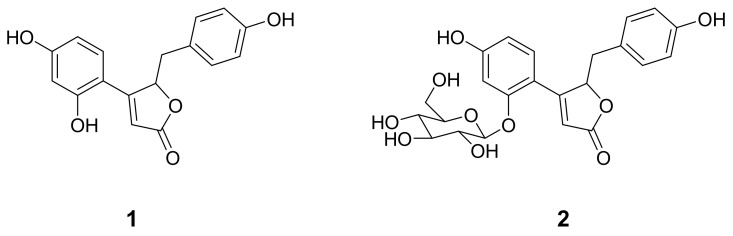
Structures of tyrosinase inhibitory compounds (**1** and **2**) from *A. fruticosa*.

**Figure 2 molecules-25-02344-f002:**
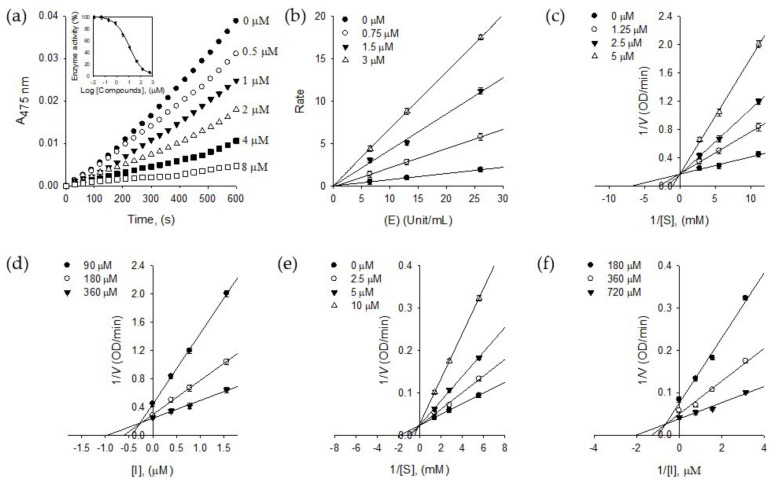
(**a**) Dose-dependent inhibitory effects of isolated compound **1** on monophenolase activity (Inset). The effect of **1** on the activity of monophenolase. (**b**) Determination of the reversible inhibitory mechanism of **1**. (**c**) Lineweaver–Burk plots for the effect of **1** on the monophenolase activity. (**d**) Dixon plots for the effect of **1** on the monophenolase activity. (**e**) Lineweaver–Burk plots for the effect of **1** on the diphenolase activity. (**f**) Dixon plots for the effect of **1** on the diphenolase activity.

**Figure 3 molecules-25-02344-f003:**
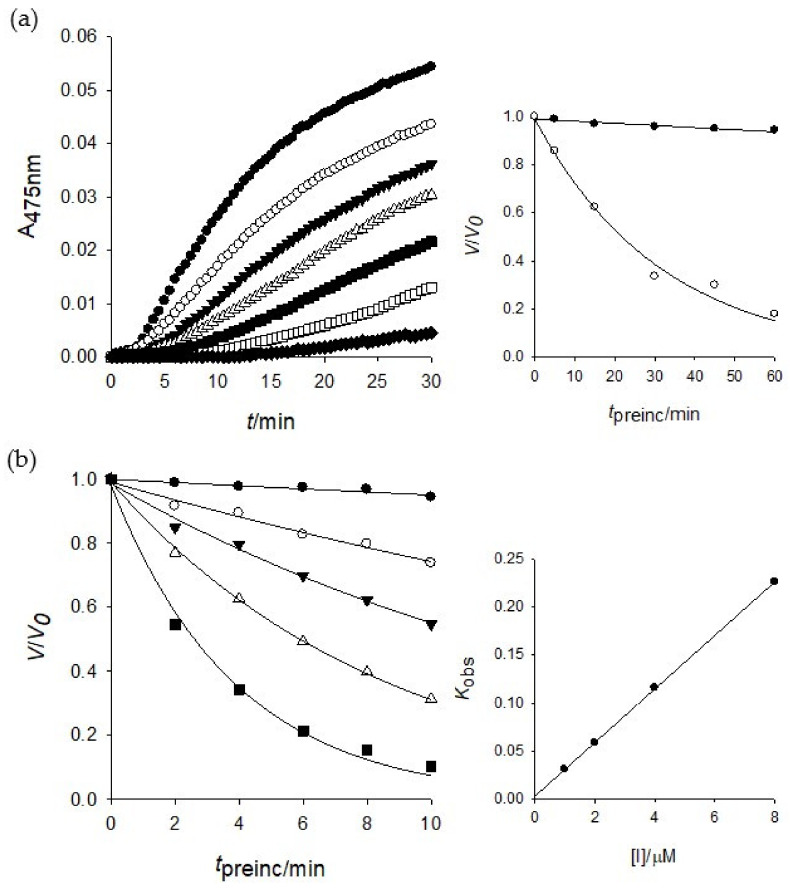
(**a**) Inhibition as a function of preincubation time (●: 0, ○: 5, ▼: 10, △: 15, ■: 30, □: 45, ♦: 60 min) for compound **1** at 2.0 μM. Inset: ●: 0, ○: 2.0 μM. (**b**) Time course of the inactivation of tyrosinase by **1** (●: 0, ○: 1.0, ▼: 2.0, △: 4.0, ■: 8.0 μM). Inset: plot of *K*_obs_ as a function of inhibitor **1** concentration.

**Figure 4 molecules-25-02344-f004:**
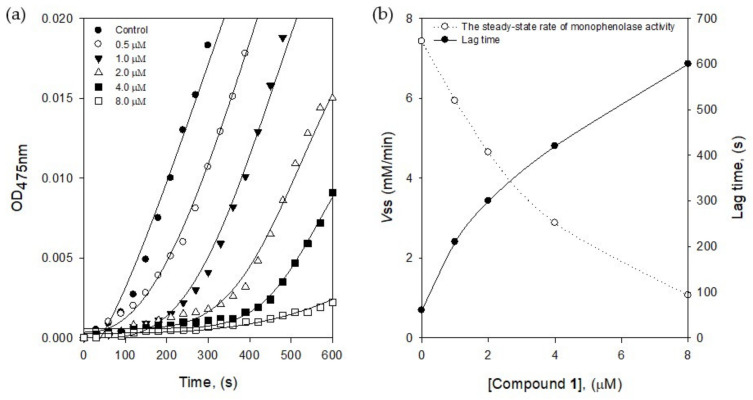
(**a**) Time course of oxidation of l-tyrosine catalyzed by tyrosinase in the presence of compound **1** at different concentrations (0, 0.5, 1.0, 2.0, 4.0, and 8.0 μM) (**b**) The steady-state rate of monophenolase activity and the lag period of monophenolase for the oxidation of l-tyrosine.

**Figure 5 molecules-25-02344-f005:**
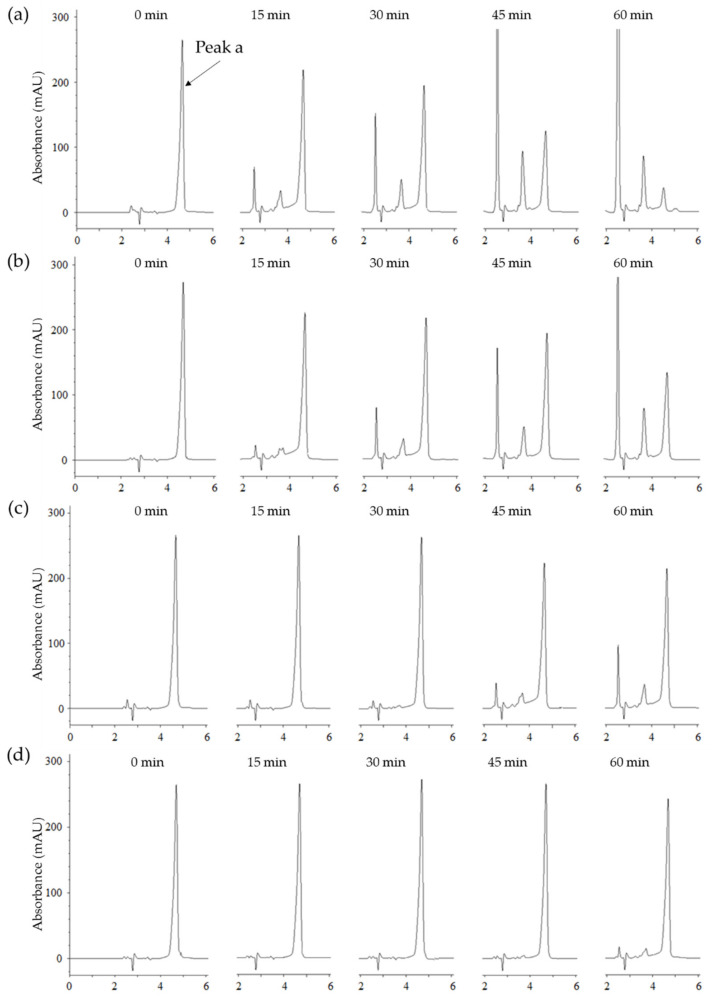
(**a**) High-performance liquid chromatography (HPLC) analysis of change in the content of *N*-acetyl-L-tyrosine (peak a, tR = 4.6 min) as a substrate for tyrosinase at different incubation times (0, 15, 30, 45, and 60 min) in the absence of compound **1**. (**b**) in the presence of 5.0 μM compound **1**. (**c**) in the presence of 10 μM compound **1**. (**d**) in the presence of 20 μM compound **1**.

**Figure 6 molecules-25-02344-f006:**
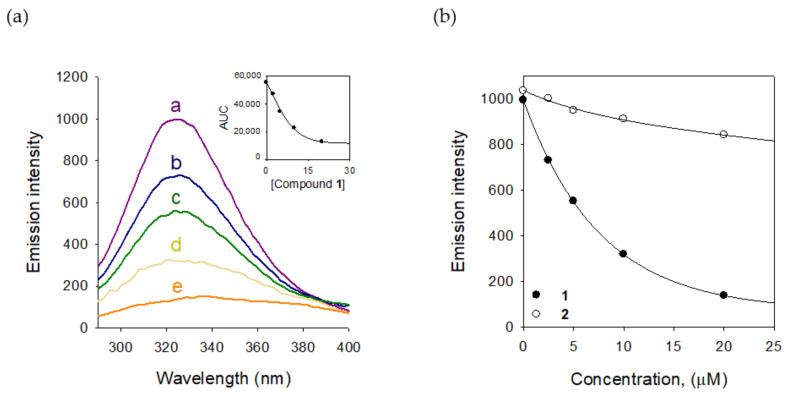
(**a**) The fluorescence emission spectra of tyrosinase at different concentrations of compound **1** (0, 2.5, 5, 10, and 20 μM for curves a→e) (Inset) Normalized intensities of the fluorescence for tyrosinase are shown for compound **1**. (**b**) Decrease in intensity of the emission plots for compounds **1** and **2**.

**Figure 7 molecules-25-02344-f007:**
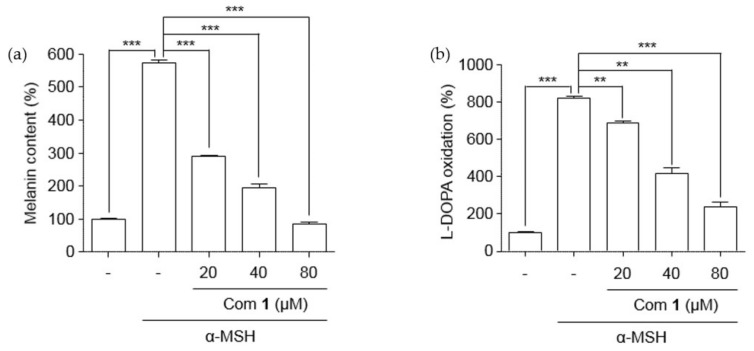
Effect of lowering (**a**) melanin content and (**b**) l-DOPA oxidation in B16F10 mouse melanoma cells. B16F10 cells were treated with different concentrations of compound **1** for 48 h in the presence of α-MSH. Data are represented as % of control, and each column indicates the mean ± standard error (SE) of the three determinations (*n* = 3); ** *p* < 0.01, *** *p* < 0.001. Compound **1** represents component **1**.

**Table 1 molecules-25-02344-t001:** Inhibitory effects of compounds (**1** and **2**) on tyrosinase inhibition activity.

Compound	l-Tyrosine		l-DOPA	
	IC_50_ ^1^ Value (μM)	Inhibition Mode (*K*_i_ ^2^, μM)	IC_50_ Value (μM)	Inhibition Mode (*K*_i_, μM)
**1**	2.20 ± 0.2	Competitive (0.87)	3.88 ± 0.3	Competitive (1.95)
**2**	>200	NT ^3^	>200	NT
Kojic acid ^4^	14.8 ± 0.6	NT	37.1 ± 1.3	NT

All compounds were tested as a set of experiments repeated three times; ^1^ IC_50_ values of compounds represent the concentration that caused 50% enzyme activity loss; ^2^ Values of inhibition constant; ^3^ NT is not tested; ^4^ Kojic acid is a positive control.

**Table 2 molecules-25-02344-t002:** Effects of compound **1** on cell growth and melanin production of B16 cells.

Compounds	Melanin Synthesis IC_50_ (μM)	l-DOPA Oxidation IC_50_ (μM)	Cytotoxicity LD_50_ (μM)
**1**	11.4 ± 1.2	23.9 ± 1.4	137.8 ± 1.2
Kojic acid ^a^	>500	>500	>500

B16 cells were grown overnight in six-well plates. Each indicated sample was added to the plate for 48 h in the presence of α-MSH. ^a^ Kojic acid was used for positive control.

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
