# Peer review of "Tyrosinase Inhibition and Kinetic Details of Puerol A Having But-2-Enolide Structure from Amorpha fruticosa"

_molecules, 2020, doi:10.3390/molecules25102344_

Round 1

Reviewer 1 Report

The manuscript shows a big amount of work and illustrates how Puerol A isolated from Amorpha fruticose can decrease the mushroom tyrosinase activity and α-MSH induced melanin overproduction in B16F10 melanoma cells. Nevertheless, there are some aspects that should be cleared up prior its consideration for publication.

  1. Puerol A is the major active ingredient of Amorpha fruticose, how did authors choose these compounds to determine the antimelanogenesis? Did authors perform the similar study with Amorpha fruticose methanol extract?
  2. How about the purity of Puerol A and Kuzubutenolide A? The HPLC chromatography of them should be added in supplementary data.
  3. In HPLC analysis of tyrosinase inhibition, no information is found concerning the HPLC peaks of N-acetyl-L-tyrosine. This aspect should be also included for the HPLC method validation.
  4. In Cell viability assay, Does the method refer to ISO 10993-5 for determining the cytotoxicity of Puerol A?
  5. In addition, why did authors use the 1μM of α-MSH to induce melanin overproduction in B16F10 melanoma cell? Other paper used 100nM and 10 nM ofα-MSH in the similar study. Why did authors not perform cellular tyrosinase activity in B16F10 cells, many studies displayed the result for elucidating the antimelanogenesis activity. Please clarify these problems.

Author Response

Dear Reviewer

Thank you very much for kind review and comments concerning our manuscript. We appreciate the hard work of reviewers as they fairly pointed out errors and mistakes in our manuscript. We tried to revise the manuscript as much as possible in line with comments made by reviewers. Corrections were colored with red in the manuscript.

Please find below the requested details according to reviewer’s comments and suggestions.

  1. Puerol A is the major active ingredient of Amorpha fruticosa, how did authors choose these compounds to determine the antimelanogenesis? Did authors perform the similar study with Amorpha fruticosa methanol extract?

→ Several kinds of enzyme screenings were applied to puerol A (1), but only mushroom tyrosinase was inhibited significantly by puerol A. Thus we tried to make a full story of puerol A against tyrosinase.

  1. How about the purity of Puerol A and Kuzubutenolide A? The HPLC chromatography of them should be added in supplementary data.

→ Yes, They might have above 98% purity based on 1H-NMR, which can apply both UV active and UV inactive compounds. We added the HPLC chromatorgram of compounds 1 and 2 to supplementary data as reviewer’s suggestion. Please see Figure. S15.

  1. In HPLC analysis of tyrosinase inhibition, no information is found concerning the HPLC peaks of N-acetyl-L-tyrosine. This aspect should be also included for the HPLC method validation.

→ Yes, there was an unclear description for reader to understand easily. We resharpened a whole description with the concrete HPLC conditions. Please see Line 424-431 in experimental section.

  1. In Cell viability assay, Does the method refer to ISO 10993-5 for determining the cytotoxicity of Puerol A?

→ Yes, we followed ISO 10993-5:2009(E) Annex C (MTT cytotoxicity test) for determining in vitro cytotoxicity of Puerol A on B16F10 cells. In addition, We added a reference, Lee K.W., et al. (2016), to the experimental section. Please see Line 457-463.

  1. In addition, why did authors use the 1μM of α-MSH to induce melanin overproduction in B16F10 melanoma cell? Other paper used 100 nM and 10 nM of α-MSH in the similar study. Why did authors not perform cellular tyrosinase activity in B16F10 cells, many studies displayed the result for elucidating the antimelanogenesis activity. Please clarify these problems

→ As reviewer told us, we set the experiment, α-MSH-mediated melanogenesis, with 1 μM α-MSH. The concentration of 1 μM α-MSH did not show any cytotoxicity and induced a certain melanin synthesis regardless of the production company and batch. Additionally, we could find several reports showing 1 μM α-MSH-mediated melanogenesis (Pharmacogn Mag. 2014 Aug;10(Suppl 3):S463-71.; MOLECULAR MEDICINE REPORTS 2017. 16: 1079-1086..; Biol. Pharm. Bull. 2012. 35(1) 78—83.; Biochemical Pharmacology. 2019. 164 299–310.)

→ In this study, we presented L-DOPA oxidation (%) (Figure. 7b) as a parameter for tyrosinase activity in the cell. As the figure 7, compound 1 significantly and dose-dependently inhibited melanin content and L-DOPA oxidation in B16F10 cells. Based on tyrosinase inhibition experiments and the result of Figure 7, we suggested compound 1 induces anti-melanogenesis through tyrosinase inhibition.

Reviewer 2 Report

The submitted paper describes tyrosinase inhibitory activity of puerol A having but-2-enolide structure from Amorpha fruticosa. Moreover, the details of the inhibitory mechanism were presented. In my opinion the study is valuable and the presented results are convincing. The paper is concise and well written.

There are only two minor points which should be addressed by the authors:

- Introduction, line 48: ‘antimicrobial activities’ vs. ‘antimicrobial inhibitory activities’

- line 275: ‘did not’ vs. ‘didn’t’

Author Response

Dear Reviewer

Thank you very much for kind review and comments concerning our manuscript. We appreciate the hard work of reviewers as they fairly pointed out errors and mistakes in our manuscript. We tried to revise the manuscript as much as possible in line with comments made by reviewers. Corrections were colored with red in the manuscript.

Please find below the requested details according to reviewer’s comments and suggestions.

  1. Introduction, line 48: ‘antimicrobial activities’ vs. ‘antimicrobial inhibitory activities’

→ ‘antimicrobial inhibitory activities’ in text was changed to ‘antimicrobial activities’ in Line 49.

  1. Line 275: ‘did not’ vs. ‘didn’t’

→ ‘didn’t’ in text was changed to ‘did not’ in Line 278

Reviewer 3 Report

Dear Authors,

the overall topic of your paper is interesting, especially taking into consideration a growing interest in identyfying novel tyrosinase inhibitors from natural sources. Please find below a few comments which I believe will help you to improve the quality of your manuscript:

 Introduction:

Please explain in more details (2-3 sentences) the biological importance of your study – the sentence in line 29 “…excess melanin may affect conditions of skin, brain and fruit” is very general and might be confusing for some readers

Line 45: Leguminosae should be written in italics

Tyrosinase inhibition:

Table 2: In my opinion the data regarding the inhibitory activity of compound 2 (percent inhibition for 2 mM compound 2) should be excluded from the table with IC50 as this is confusing. It would be better to mention this values in the text and also discuss the tyrosinase inhibitory properties of compound 2 - could you suggest the possible explanation of low tyrosinase inhibitory activity of compound 2, despide the structural similarity with compound 1?

Binding affinity to tyrosinase and antipigmentation

I would suggest to divide this section into two, for example “Binding affinity to tyrosinase” and “Antipigmentation activity in vitro”. Due to the structural and some functional differences between mushroom and mammalian tyrosinases I think it is important to clarify the origin of tyrosinase used in the binding assay. Based on the structure of mammalian/human tyrosinase is it possible to predict if the binding affinity of compound 1 to the mammalian enzyme will be similar?

3.3 Inhibitory effects against tyrosinase

Please provide the source of tyrosinase enzyme used in this assay (origin and manufacturer)

3.5 HPLC analysis of tyrosinase inhibition

Please provide the source of tyrosinase enzyme used in this assay (origin and manufacturer) and its activity (U/mL).

3.8 Cell viability assay

Please provide the initial number of cells plated onto wells.

Line 456 “…the cells were allowed to stand for 3 hours” please change for “the cells were incubated for 3 hours”

3.9 Measurement of melanin content

Please provide the initial number of cells plated onto wells.

3.10 Measurement of L-DOPA oxidation

Please provide the initial number of cells plated onto wells.

Line 473: “The proteins were cultured for 4 hours with 100 µl…” – this sentence does not make sense, please correct the description of the procedure.

As I understand, in procedures described in 3.3 and 3.10 the product of L-DOPA oxidation by tyrosinase was measured, using different source of tyrosinase enzyme. Could you please explain why different wavelengths were used for these analyses (475 nm in procedure 3.3 and 490 nm in procedure 3.10)?

Author Response

Dear Reviewer

Thank you very much for kind review and comments concerning our manuscript. We appreciate the hard work of reviewers as they fairly pointed out errors and mistakes in our manuscript. We tried to revise the manuscript as much as possible in line with comments made by reviewers. Corrections were colored with red in the manuscript.

Please find below the requested details according to reviewer’s comments and suggestions.

  1. Please explain in more details (2-3 sentences) the biological importance of your study – the sentence in line 29 “…excess melanin may affect conditions of skin, brain and fruit” is very general and might be confusing for some readers

→ As reviewer’s suggestion, description was adjusted to focus on skin and neurotoxicity. Please see Line 29-33.

  1. Line 45: Leguminosae should be written in italics

→ “Leguminosae” in text was changed to “Leguminosae” in italics. Please see Line 45.

  1. Table 1: In my opinion the data regarding the inhibitory activity of compound 2 (percent inhibition for 2 mM compound 2) should be excluded from the table with IC50 as this is confusing. It would be better to mention this values in the text and also discuss the tyrosinase inhibitory properties of compound 2 - could you suggest the possible explanation of low tyrosinase inhibitory activity of compound 2, despide the structural similarity with compound 1?

→ Table 1 was resharpen with a sign of inequality (> 200 μΜ) to be with smooth story. Please see Table 1. The reason that compound 2 showed the low tyrosinase inhibitory activity was the loss of resorcinol group, which is well-known chemical moiety that inhibits tyrosinase. We also mentioned it in the text. Please see Line 83-85.

  1. Binding affinity to tyrosinase and antipigmentation : I would suggest to divide this section into two, for example “Binding affinity to tyrosinase” and “Antipigmentation activity in vitro”. Due to the structural and some functional differences between mushroom and mammalian tyrosinases I think it is important to clarify the origin of tyrosinase used in the binding assay. Based on the structure of mammalian/human tyrosinase is it possible to predict if the binding affinity of compound 1 to the mammalian enzyme will be similar?

→ As reviewer’s suggestion, the subtitle was divided to “Binding affinity to tyrosinase” and “Antipigmentation activity in vitro”. Please see each subtitle in manuscript. Of course, both tyrosinase from mushroom (Agaricus bisporus) and human have a very high similarity (Panzella, L.; Napolitano, A., 2019). Thus, compound 1 has a possibility showing a comparable binding affinity to the human enzyme. 

  1. Inhibitory effects against tyrosinase : Please provide the source of tyrosinase enzyme used in this assay (origin and manufacturer)

→ We provided the source of tyrosinase to experimental section. Please see Line 387.

  1. HPLC analysis of tyrosinase inhibition : Please provide the source of tyrosinase enzyme used in this assay (origin and manufacturer) and its activity (U/mL).

→ We provided a detail information of enzyme to experimental section. Please see Line 426.

  1. Cell viability assay : Please provide the initial number of cells plated onto wells. Line 456 “…the cells were allowed to stand for 3 hours” please change for “the cells were incubated for 3 hours.

→ “…the cells were allowed to stand for 3 hours” was changed to “the cells were incubated for 3 hours.” Please see Line 460.

  1. Measurement of melanin content : Please provide the initial number of cells plated onto wells.

→ We added the initial cell number at the experimental section. Please see Line 466-467.

  1. Measurement of L-DOPA oxidation : Please provide the initial number of cells plated onto wells.

→ We added the initial cell number at the experimental section. Please see Line 474-475.

  1. Line 473: “The proteins were cultured for 4 hours with 100 µL…” – this sentence does not make sense, please correct the description of the procedure.

→ Line 478: “The proteins were cultured for 4 hours with 100 µl…” was changed to “Then, the cells were cultured for 4 hours with 100 μL…”.

  1. As I understand, in procedures described in 3.3 and 3.10 the product of L-DOPA oxidation by tyrosinase was measured, using different source of tyrosinase enzyme. Could you please explain why different wavelengths were used for these analyses (475 nm in procedure 3.3 and 490 nm in procedure 3.10)?

→ In materials and methods section 3.3, purified tyrosinase from mushroom was used for L-DOPA oxidation and we followed ‘Journal of Toxicology and Environmental Health, Part A, 70(5), 393–407’. In this paper 475 nm wavelength was used. But in section 3.10, we followed the method by Choi et al (Food Funct. 2013 Oct;4(10):1481-8’) to determine L-Dopa oxidation in cells. We changed Ref # 25 with the paper published at Food. Fuct. 2013